# An Emerging Cross-Species Marker for Organismal Health: Tryptophan-Kynurenine Pathway

**DOI:** 10.3390/ijms23116300

**Published:** 2022-06-04

**Authors:** Laiba Jamshed, Amrita Debnath, Shanza Jamshed, Jade V. Wish, Jason C. Raine, Gregg T. Tomy, Philippe J. Thomas, Alison C. Holloway

**Affiliations:** 1Department of Obstetrics and Gynecology, McMaster University, Hamilton, ON L8S 4K1, Canada; jamshel@mcmaster.ca (L.J.); debnata@mcmaster.ca (A.D.); sjamshed@uwaterloo.ca (S.J.); 2Department of Chemistry, Centre for Oil and Gas Research and Development (COGRAD), University of Manitoba, 586 Parker Building, 144 Dysart Rd., Winnipeg, MB R3T 2N2, Canada; wishj@myumanitoba.ca (J.V.W.); gregg.tomy@umanitoba.ca (G.T.T.); 3Quesnel River Research Centre, University of Northern British Columbia, Prince George, BC V2N 4Z9, Canada; jason.raine@unbc.ca; 4Environment and Climate Change Canada, National Wildlife Research Centre, Ottawa, ON K1A 0H3, Canada; philippe.thomas@ec.gc.ca

**Keywords:** tryptophan, kynurenine, metabolites, glucocorticoids, stress, inflammation, oxidative stress, toxicants, environmental contaminants, biomarker

## Abstract

Tryptophan (TRP) is an essential dietary amino acid that, unless otherwise committed to protein synthesis, undergoes metabolism via the Tryptophan-Kynurenine (TRP-KYN) pathway in vertebrate organisms. TRP and its metabolites have key roles in diverse physiological processes including cell growth and maintenance, immunity, disease states and the coordination of adaptive responses to environmental and dietary cues. Changes in TRP metabolism can alter the availability of TRP for protein and serotonin biosynthesis as well as alter levels of the immune-active KYN pathway metabolites. There is now considerable evidence which has shown that the TRP-KYN pathway can be influenced by various stressors including glucocorticoids (marker of chronic stress), infection, inflammation and oxidative stress, and environmental toxicants. While there is little known regarding the role of TRP metabolism following exposure to environmental contaminants, there is evidence of linkages between chemically induced metabolic perturbations and altered TRP enzymes and KYN metabolites. Moreover, the TRP-KYN pathway is conserved across vertebrate species and can be influenced by exposure to xenobiotics, therefore, understanding how this pathway is regulated may have broader implications for environmental and wildlife toxicology. The goal of this narrative review is to (1) identify key pathways affecting Trp-Kyn metabolism in vertebrates and (2) highlight consequences of altered tryptophan metabolism in mammals, birds, amphibians, and fish. We discuss current literature available across species, highlight gaps in the current state of knowledge, and further postulate that the kynurenine to tryptophan ratio can be used as a novel biomarker for assessing organismal and, more broadly, ecosystem health.

## 1. Introduction

The aromatic amino acid L-tryptophan (TRP) is one of nine essential amino acids required by all living organisms for protein synthesis. Although bacteria, fungi and plants can synthesize TRP from phosphoenolpyruvic acid via the shikimate pathway [1], vertebrates must obtain TRP from dietary sources [2,3], such as plants and plant products (i.e., fruit, nuts, oats, chocolate) and other animals and animal products (i.e., fish, turkey, dairy). In addition to its role in protein synthesis, TRP is the precursor of the monoaminergic neurotransmitter, serotonin (5-hydroxytryptamine, 5-HT) [2,3,4,5]. 5-HT is known to regulate behaviour and adaptive responses towards environmental stress. These responses include mood (anxiety) [6,7,8], cognition [9,10], nociception, aggressiveness, appetite [11,12], and temperature homeostasis [13]. Furthermore, there is evidence that 5-HT and its derivative melatonin regulate reproductive processes in vertebrates through the hypothalamo-hypophyseal system and by direct effects on reproductive organs [14,15,16]. Although the role of TRP and its metabolites may differ among mammals, birds, and fish, all vertebrates produce 5-HT in the central nervous system [4,5,17,18,19]. More recently, peripheral 5-HT synthesis and signaling has been identified in mammals [20,21] and fish [15,22]. Peripheral 5-HT has been reported to modulate gonadal hormone secretion, immune and inflammatory responses, and metabolic homeostasis in the gut [4,23]. Although TRP is a critical precursor for serotonin synthesis, only 1–2% of dietary TRP is converted to 5-HT; approximately 95% of dietary TRP is metabolized through the tryptophan-kynurenine (TRP-KYN) pathway [3].

### 1.1. Tryptophan Metabolites

The kynurenine (KYN) pathway of TRP metabolism produces biologically active metabolites involved in inflammation, immune responses, and excitatory neurotransmission [3,24,25]. The first step involves the oxidation of the indole ring and is catalyzed by two enzymes that are differentially distributed: tryptophan 2,3-dioxygenase (TDO) (predominantly expressed in the liver and brain) and indoleamine 2,3-dioxygenease (IDO) (expressed in most peripheral tissues) [3,26,27] (Figure 1). These heme-containing enzymes are structurally distinct proteins that have evolved to catalyze the conversion of TRP to *N*-formylkynurenine. *N*-formylkynurenine is rapidly converted to KYN by *N*-formylkynurenine formamidase. At this point, KYN metabolism can follow one of two branches: (1) it can be metabolized to kynurenic acid (KYNA) and anthranilic acid (AA) via kynurenine aminotransferase (KAT) and kynureninase (KYNU), respectively, or (2) it can be converted to the neurotoxic and free-radical generator 3-hydroxykynurenine (3-HK), which is further transaminated to xanthurenic acid (XA). KYNU catalyzes the hydrolysis of 3-HK to form the immunomodulatory and free-radical generator 3-hydroxyanthranilic acid (3-HAA), which can spontaneously transform to nicotinic acid (NIC) via 3-hydroxyanthranilate 3,4-dioxygenase (3-HAAO) or be completely oxidized to carbon dioxide (CO_2_). Initially, 3-HAAO converts 3-HAA to the unstable intermediate 2-amino-3-carboxymuconic semialdehyde (ACMS), which spontaneously rearranges to the excitotoxin and *N*-methyl-D-aspartate (NMDA) receptor agonist quinolinic acid (QUIN) or is further converted to the neuroprotective picolinic acid (PIC) after enzymatic decarboxylation via 2-amino-3-carboxymuconate semialdehyde decarboxylase (ACMSD). QUIN is decarboxylated to form NIC, the NAD^+^ coenzyme precursor. QUIN is also the substrate for quinolinate phosphoribosyltransferase (QPRT), which initiates several metabolic steps to ultimately produce the essential cofactor, NAD^+^ [3,4,25].

### 1.2. Kynurenine Metabolites as Biomarkers of Human Disease

TRP and its metabolites have key roles in diverse physiological processes such as cell growth and maintenance (where TRP serves as a building block for proteins) and the coordination of adaptive responses to environmental and dietary cues (where TRP metabolites serve as neurotransmitters and signaling molecules). For example, many KYN metabolites are neuroactive and are considered cytoprotective (such as KYNA) or cytotoxic (such as 3-HK, 3-HAA, QUIN, and NIC) [3,28]. These effects are highly dependent on the accumulation of metabolites within specific tissues [Reviewed In: [29]]. KYNA acts as an antagonist to α-amino-3-hydroxy-5-methyl-4-isoxazolepropionic acid (AMPA), NMDA and kainate glutamate receptors, while QUIN acts as an NMDA receptor agonist [30,31]. NAD^+^ is an important coenzyme in many energy metabolism pathways including glycolysis, β-oxidation, and oxidative phosphorylation [32], and picolinic acid modulates immune function and antimicrobial activity [33].

Over the past 15 years, there has been increasing interest in the role of KYN metabolites in human disease models (Appendix A). Indeed, altered levels of TRP-KYN pathway metabolites have been reported in aging and sleep disorders [34,35], metabolic syndrome [23,24,36], cardiovascular disease [37,38], cancer [39], autoimmune disease [40,41,42], anxiety and depression [6,7], neurodegenerative diseases [30], as well as obesity, anorexia and bulimia nervosa, and other diseases presenting peripheral symptoms [19,42]. Changes in TRP metabolism can alter the availability of TRP for protein and 5-HT biosynthesis and dysregulate the levels of immuno- and neuro-active KYN metabolites. Recently, research has begun to assess KYN metabolites as biomarkers to many neuro- and immune-associated illnesses, including Alzheimer’s Disease [43,44,45], ALS [46], and Major Depressive Disorder [47]. 

### 1.3. Objective of Study

However, to assess KYN enzymes and metabolites as markers of altered homeostasis, it is critical to understand and appreciate the integration of TRP catabolism. KYN pathway activation, enzymatic activity, and metabolite formation and function is dependent on exogenous stimuli and endogenous ligand binding. There is now considerable evidence showing that the KYN pathway of TRP metabolism can be influenced by various stressors including glucocorticoids (GCs) [24,47,48,49,50,51], infection [52,53,54,55,56], inflammation and oxidative stress [57,58,59,60,61,62,63], and environmental contaminants [3,64,65,66].

Importantly, the TRP-KYN pathway is conserved across vertebrate species, understanding how this pathway is regulated may have broader implications for environmental and wildlife toxicology. In mammalian species and birds with monogastric systems including duck (*Anatidae*) and chicken (*Gallus gallus*), the KYN pathway is the central catabolic route for TRP’s indole ring [2,3]. As reviewed by Ball and colleagues [67], the genes of the major TRP catabolizing enzymes IDO and TDO have undergone gene duplications leading to multiple isoforms of IDO. For example, all mammals have two IDO genes (IDO1 and IDO2) via gene duplication, whereas fish and amphibians have IDO2-like genes. Interestingly, while IDO and TDO enzymes show functional convergence, TDO has a higher catalytic efficiency for TRP catabolism, and as such, its conservation across metazoans and vertebrates is clear [27]. More importantly, the expression of these enzymes differ across species by tissue/organ/cellular localization, by enzymatic characteristics, signaling properties, and by biological function following induction by distinct stimuli [27,67]. 

The goal of this narrative review is to identify (1) key pathways affecting TRP-KYN metabolism in vertebrates and (2) consequences of altered TRP metabolism in mammals, birds, amphibians, and fish. We provide evidence of studies linking exposure to environmental contaminants and altered TRP-KYN metabolites, which contribute to a range of adverse health outcomes. Furthermore, we postulate that alterations in TRP metabolism, KYN metabolites, and the TRP:KYN ratio may be an indication of exposure to environmental contaminants. Thus, we suggest that levels of KYN metabolites and the TRP:KYN ratio can be used as a novel integrative biomarker for assessing organismal exposure to environmental contaminants and, more broadly, ecosystem health.

## 2. Methods

A literature search was conducted in NCBI using MeSH on “Tryptophan” OR “Kynurenine” AND the terms: “Glucocorticoids”, “Stress, Physiological”, “Infections”, “Inflammation”, “Oxidative Stress”, “Environmental Pollution/adverse effects”, “Environmental Pollution/genetics”, “Environmental Pollution/immunology”, “Environmental Pollution/metabolism”, “Environmental Pollution/pathogenicity”, “Environmental Pollution/pathology”, “Environmental Pollution/physiology”, “Environmental Pollution/physiopathology”, and “Environmental Pollution/toxicity”. Collectively, between the years 1970–2021, a total of 1959 publications were identified. From these, we selected publications that specifically identified changes in key cellular/physiological pathways that led to alterations in TRP-KYN metabolites in vertebrates (i.e., mammals, birds, fish, amphibians), and publications that identified altered tryptophan metabolites and their physiological consequences. A total of 265 papers were selected, and of those 185 were used in this review to focus on (1) glucocorticoids and stress, (2) infection, inflammation, and oxidative stress and (3) environmental contaminants. The break down by species and by physiological processes/pathways is depicted in Figure 2.

An additional screening was completed to identify enzyme and metabolite presence across vertebrate animal kingdoms (Figure 3). Both gene detection (i.e., PCR of mRNA, RNA, and DNA) and protein levels (i.e., HPLC, LC, MS, Western Blots) were methodologically screened using literature searches and databases. Gene data was collected through PubMed and NCBI searches specified for “gene name” (i.e., IDO, TDO, TPH, KYN, KYNA etc.) AND “homo sapiens” OR for mammals [“mammal”, “rat”, “mouse”, “rabbit”] OR for fish [“fish”, “medaka”, “salmon”, “trout”, “carp”] OR for birds [“bird”, “pigeon”, “chicken”], OR for amphibians [“amphibian”, “frog”]. It is important to note that when screening for enzyme and gene names, all isoforms of IDO and KAT were identified and used as “gene name”. If genomic information was present, it was considered “detected”. For protein levels, the UNIPROT database was screened with “kingdom” (i.e., mammal, bird, fish, frog, amphibian) and “Enzyme Name”. Entries were then confirmed by identifying protein name, organism, and whether or not it was reviewed (Swiss-Prot) or unreviewed (TrEMBL, computational curation). If it was reviewed, it was considered “detected”. If it was unreviewed, a PubMed and NCBI check was conducted, using MeSH terms for “Enzyme Name”, selecting “Other Animals” and a species name if needed (e.g., Gallus gallus). If an enzyme was unreviewed, and PubMed and NCBI identified primary articles that showed protein levels of the enzyme, it was considered “detected”. If an enzyme was unreviewed and no additional information could be found, it was considered “orthologs may be present but have not been confirmed”. If no information was found, it was considered “unknown”. Metabolites could not be screened using UNIPROT, so literature data was used primarily to identify the presence across animal kingdoms. 

## 3. Discussion

In the following sections, we discuss critical stress-inducing pathways that may play a role in driving TRP metabolism down the KYN pathway. As the TRP-KYN pathway is altered in response to glucocorticoids and chronic stress, infection, inflammation, and oxidative stress, and environmental toxicants, and these stressors are ubiquitous phenomenon across animal kingdoms, we explore the current literature available with regard to mammals, birds, fish, and amphibians. 

### 3.1. Glucocorticoids and Chronic Stress

Glucocorticoids (GCs) are hormones of the endocrine system responsible for regulating stress responses. GCs are modulated by the hypothalamic-pituitary-adrenal (HPA) axis through the release of tropic hormones and negative feedback regulation. Although GCs exhibit circadian variation, they can be induced in response to physical, chemical, and psychological stressors. The molecular mechanism begins with the binding of GCs to the nuclear glucocorticoid receptor (GR), which functions as a transcription factor and modulates downstream gene expression. In general, increased GC signaling has anti-inflammatory and immunosuppressive activity. When responding to stress, the body mobilizes energy through sympathetic activation, which governs the “fight-or-flight” response. For this reason, the stress response sits at the interface of neuroendocrine, immune, and behavioural systems.

The degree of the stress response often depends on the duration and intensity of the stressor. Acute, immediate stress responses tend to be protective and allow an organism to adapt to a changing environment [63,68]. Conversely, chronic, prolonged stress activation can lead to the accumulation of wear-and-tear on physiological systems and result in impaired health outcomes. For this reason, it is important to manage the balance between protective and damaging stress responses. In environmental toxicology, biomarkers such as metabolic cytochrome P450 (CYP) enzymes and heat shock proteins are often used as indicators for exposure to acute stressors and environmental insults [69]. Not only are these markers sensitive to contaminants and inexpensive for use in field monitoring, they are reliable in a wide variety of organisms. While acute stress-induced GC increase in plasma is associated with increased 5-HT production through TRP hydroxylase activity, chronic stress reduces 5-HT turnover and release and favours the KYN pathway [7]. Given the role of the TRP-KYN pathway in stress responses, its metabolites and enzymes may be useful biomarkers for stress across species. 

It is generally accepted that GCs increase the expression and activity of hepatic TDO in rats and mice [29,70,71,72,73]. GC signaling involves de novo synthesis of TDO at the transcriptional level, which is mediated by GC-responsive elements on the promoter region of TDO [72,73]. TDO activity can be increased through substrate activation by TRP and cofactor activation by heme, neither of which involve de novo synthesis. Chronic stress responses, which involve both HPA signaling and sympathetic activation, increase the metabolism of TRP to KYN by TDO, which reduces the availability and transport of TRP to the brain [3] for 5-HT production. While IDO expression is not directly induced by GCs, HPA tropic hormones and/or stress can increase the expression and activity of IDO. This has been shown in the hippocampus of mice treated with adrenocorticotropic hormone [74], and in the serum and frontal cortex of rats exposed to chronic mild stress [75,76]. Some studies have suggested that TRP uptake into the brain is increased with acute stress and HPA activation. For example, in rats, foot shock stress increased levels of TRP, KYN, KYNA, and 3-HK in the brain. Similarly, acute stress induced by physical restraint in mice increased TRP metabolism to KYN in the brain and plasma, along with an increase in IDO1, IDO2, and TDO2 [77]. Moreover, following exposure to a novel stressor, brain TRP and KYN levels were raised in mice with no change in the TRP:KYN ratio and a reduction in the 5-HT:TRP ratio [78]. This suggests that while brain TRP levels increase with stress, brain IDO activity may also be induced, which increases KYN levels and shifts metabolism away from the 5-HT pathway [5].

Stress-induced alterations in TRP metabolism in the brain and periphery in mammals can also be due to an interaction between the immune system and the HPA axis [78]. Exposure to stressors induces the release of proinflammatory cytokines, which further activate the HPA axis. For example, proinflammatory interleukin-1 (IL-1) is a potent stimulant for corticotropin releasing hormone (CRH) synthesis. By increasing CRH levels, GC production can be increased. Additionally, the proinflammatory cytokine interferon-gamma (IFN-γ) induces IDO expression in hypothalamic and pituitary neurons, which produce 3-HK and QUIN [78]. Induction of IDO by IFN-γ can be potentiated by dexamethasone, which has no effect when administered alone in human monocytes [79]. In mice, stimulation of the HPA axis through immune activation by IL-1 and lipopolysaccharide has been shown to increase brain TRP levels. However, since stress-induced changes in brain TRP levels occur in adrenalectomized rats and mice, this is not necessarily dependent on adrenocortical activation [78]. Similarly, Rose et al. (2020) reported increased KYN and decreased TRP levels in plasma, and decreased expression of monoamine oxidase A (MAOA) following exposure to ozone; the addition of metyrapone, a GC inhibitor, was able to reverse the effects of ozone on MAOA and kynurenine monooxygenase (KMO), but did not change the effects on TRP and KYN levels [80]. Interestingly, it has been shown that GC release is inhibited with increased KYNA levels in some areas of the central nervous system [81]. Furthermore, the enol tautomer of indole-3-pyruvic acid (IPA), another metabolite in the TRP-KYN pathway, has been reported to reduce plasma GC levels and GR activity in the hippocampus following repeated stress [81]. Taken together, these data suggest that stress plays a key role in mammalian TRP-KYN metabolism through both GC-dependent and -independent mechanisms. 

There is little known regarding the molecular effects of stress on the regulation of the TRP-KYN pathway in birds. However, in avian species, the TRP-KYN metabolic pathway is associated with undesirable behavioural patterns. For example, when chickens were under socially disruptive stress due to housing changes and manual restraint, it was found that increased destructive feather-pecking behaviour was associated with decreased plasma TRP:KYN ratios. In the same study, the socially disrupted group had gained more weight. Since plasma metabolite levels can vary with food intake, and stress can either increase or decrease feeding behaviours, it is unclear whether the observed results were due to a stress response or increased food intake [82]. Similar to rats, it was found that chickens with a TRP-supplemented diet showed reduced aggressive behaviour such as feather pecking; however, it is not clear if this is due to alterations in TRP-KYN pathway metabolites or increased 5-HT production [82]. 

In fish, stress responses involve the release of catecholamines from chromaffin tissue in the head, kidney and adrenergic nerves, as well as cortisol from the interrenal tissue into the circulation. Chronic, prolonged stress responses can lead to metabolic dysfunction and immunosuppression, which are characteristic of HPA hyperactivity [83]. The majority of work in fish has focused on the impact of TRP-supplemented diets on stress response reduction [84,85,86,87,88,89]. For example, studies in meagre (*Argyosomus regius*), rainbow trout (*Oncorhynchus mykiss*), Sengalese soles (*Solea sengalensis kaup*), Atlantic salmon (*Salmo salar*) and Atlantic cod (*Gadus morhua*) species have investigated the effects of a TRP-supplemented diet and chronic stress on TRP-KYN metabolism. Chronic stress in fish fed a control diet did not affect brain levels of 5-HT or TRP concentrations, but liver TRP, KYN, and QUIN were increased; in fish given a TRP-supplemented diet, brain 5-HT and liver TRP were reduced, and hepatic KYN and QUIN were increased [84]. Similarly, Wish et al. (2022) have found increased KYN levels in the brain and liver of rainbow trout following exposure to an acute stressor [90]. Taken together, this suggests that the KYN pathway is enhanced in stressed fish, regardless of diet. 

Across species, it appears that mammals, birds, and fish have similar responses to stress with respect to changes in TRP-KYN metabolism (Figure 4). Increased GC and catecholamine signaling, which are characteristic of stress responses, elevate TRP metabolism through the KYN pathway, as indicated by the levels of downstream metabolites. While it is generally accepted that this occurs through hepatic TDO induction in mammals, the molecular mechanism is not yet clear in other vertebrate species. In bird and fish species, TRP-enhanced diets have been shown to reduce stress-induced plasma GC levels and aggressive behaviours.

### 3.2. Infection, Inflammation and Oxidative Stress

In all species, there is an interaction between the neuroendocrine and immune systems. Both immune and endocrine cells share common receptors, and different hormones and cytokines are involved in many of the same physiological processes. TRP functions similarly in all higher vertebrates to regulate (1) the activation, proliferation, and migration of immune-surveillant cells (i.e., T- and B-lymphocytes, macrophages, and natural killer cells) and (2) the production of inflammatory signaling molecules, cytokines, nitric oxides and superoxides (Figure 5). In mammals, KYN metabolites are reported to be involved in inflammation, immune response, and excitatory neurotransmission [24]. In recent years, metabolites including KYN, KYNA, and QUIN are emerging as key targets in diseases such as diabetes, HIV [91], atherosclerosis [92], neurodegenerative diseases including schizophrenia, Alzheimer’s and Huntington’s [93,94,95], and cancer [24,96]. Primarily, these pathologies converge on inflammatory events which include target organ infiltration of circulating immune cells, activation of pro-inflammatory signaling pathways, expression of cytokines [24]/chemokines [97], and the production of reactive oxygen species (ROS) [92]. Recent studies have reported that TRP deficiencies result in immunosuppression and a significant increase in the susceptibility of humans [60], pigs, and teleost fish [98] to disease, infection, morbidity, and mortality. As such, the TRP-KYN pathway is recognized as an important player in inflammation and immune response [94,99].

Of the rate-limiting enzymes in the TRP-KYN pathway, the IDO enzymes are known to play a role in modulating infection and autoimmunity [100]. In fact, the biological role of IDO on the infiltration of circulating immune cells has been coined the TRP-depletion hypothesis; IDO suppresses microbial infection by reducing TRP availability in infected tissues [101,102]. Furthermore, IDO is highly regulated by superoxides and inflammatory mediators, where its expression is increased by IFN-γ[102,103,104,105], tumor necrosis factor-alpha (TNFα) [105,106], tumour growth factor-beta (TGF-β) [96], interleukins 1-beta, 2, 12, 18 (IL1β, IL2, IL12, and IL18) [96], and prostaglandins (PGE2) [105,107,108], as well as pathogenic infections including parasites, viruses, and bacteria. Downstream of IDO activation, the metabolite KYNA promotes monocyte extravasation and IL6 production and controls cytokine release [24,109], while the metabolite 3-HAA can induce apoptosis in T-cells through glutathione depletion [110]. In line with this, high KYN levels can increase the proliferation and migratory capacity of cancer cells and aid tumours in evading immune surveillance. In addition, KYN metabolites can then act via the aryl hydrocarbon receptor (AhR) to mediate T-cell anergy and apoptosis, proliferation of T-regulatory and T-helper(Th)17 cells, and the Th1/Th2 response [100]. Similarly, KYN metabolite KYNA has also been found to activate human, mouse, and rat GPR35—a G protein-couple receptor with orthologs found in a variety of mammals and one amphibian species (*Xenopus tropicalis)*. While KYNA has been found to be an endogenous activator of GPR35 in humans and rats altering immune responses in inflammation, pain, cancer, cardiovascular disease and energy homeostasis [Reviewed In: [111]], its role as an activator of GPR35 in other species remains unknown. 

There is increasing evidence that IDO’s role in immune function may begin in gestation. Histochemical studies of the human decidua have reported alterations in the expression of IDO and KMO throughout pregnancy. In the first trimester, IDO and KMO expression is present in stromal and glandular epithelial cells of the decidua. In both the first and second trimester placenta, they become localized to the syncytiotrophoblast, stroma, and macrophages before shifting to fetal endothelial cells and macrophages in terminal villi of term placenta [112,113,114,115]. This shift in expression suggests that the function of these enzymes may change from a role of immunosuppression at the maternal-fetal interface in early pregnancy to one associated with the regulation of fetoplacental blood flow or placental metabolism in late gestation [112]. Work done by Williams and colleagues found that 24 h following inflammation induced by intrauterine endotoxin administration, there was a significant upregulation of IDO in the placenta and fetal brain, which was associated with increased IFN-γ expression and increased in KYN, KYNA and QUIN levels. These increases occurred in parallel with decreased levels of 5-hydroxyindole acetic acid, a precursor for 5-HT [105]. Taken together, these results indicate that maternal inflammation can shunt TRP metabolism away from 5-HT and down the KYN pathway. 

IFN-γ also upregulates other enzymes in the KYN pathway, including KMO, KYNU, and 3-HAAO activities [116]. For example, in adipose tissues, which have resident macrophages with high levels of KMO, the catabolism of TRP via KMO induction following immune system activation leads to the increased production of QUIN rather than KYNA, which is reported to have anti-inflammatory properties [117]. Interestingly, plasma neopterin levels, an indicator of IFN-γ activity, have been paralleled by increased levels of the TRP:KYN ratio, which is reflective of IDO activity in patients with metabolic disorders such as type 2 diabetes [118] and neurodegenerative disorders such as Huntington’s Disease [95]. In fact, in Huntington’s patients, the TRP:KYN ratio was higher and associated with elevated levels of C-reactive protein, neopterin and lipid peroxidation products. Increased plasma KYN levels and TRP:KYN ratios have been found in patients with systemic inflammatory response syndrome, sepsis and septic shock, but the biological significance and prognostic value of these findings have remained uncertain [119]. Together, TRP:KYN and plasma neopterin levels are considered systemic markers of inflammation and oxidative stress [92,95]. 

Similar to mammalian IFN-γ, chicken IFN-γ is a known regulator of the immune response through the regulation of proinflammatory cytokines that correlate to immune activation and antiviral status. In fact, studies have previously shown that immunocompromised chickens have downregulated IFN-γ [120], which resulted in higher mortality following viral infection than normal chickens. Yuk and colleagues (2016) showed that siRNA suppression of IFN-γ in chicken embryo fibroblasts results in increased replication of viral genes during infection [121]. Taken together, these studies suggest that the immune state of chickens and other avian species are mediated by interferons [122,123]. Interestingly, Emadi and colleagues (2010) have shown that supplementing broiler chicken feed with double the National Research Council’s recommended levels of TRP enhanced the IFN-α, IFN-γ, and immunoglobulin G response to infection [124]. Taken together, these studies show that TRP may also be playing a vital role in the immunity of avian species. 

In fish, TRP plays a critical role in macrophage and lymphocyte function. When faced with infection, TRP and IDO levels are related to the induction of anti-inflammatory signaling molecules, the main effectors being T-cells (Reviewed in: [125,126]). Similar to the other vertebrates discussed, interferons that regulate IDO play analogous antiviral roles in teleost fish (Reviewed in: [127,128]). Notably, fish interferons possess the same exon/intron structure as mammalian IFN-γ [127], also playing a similar critical role in adaptive cell-mediated immune responses produced by Th1 and cytotoxic T-lymphocytes [127,129]. Interestingly, TRP supplementation has not shown improvement in immune status in European seabass (*Dicentrarchus labrax*) [130] or Persian sturgeon (*Acipenser persicus*) [131]. Furthermore, Machado et al. (2015) have shown that following TRP supplementation and subsequent increases in cortisol, European seabass had decreased monocyte/macrophage activation in response to infection. The decreased levels of lymphocytes and monocytes/macrophages were accompanied with decreased mRNA expression of proinflammatory cytokines (IL1β, IL8 and TGFβ), diminished IFN-γ, and lowered levels of superoxide dismutase (SOD), an enzyme involved in antioxidant defence [130]. Conversely, in mice [132] and broiler chickens [124], dietary TRP supplementation was able to alleviate inflammatory responses by attenuating the migration of inflammatory cells. While an interesting difference amongst vertebrates, it has been suggested that these differences may be a result of teleost fish requiring an optimal amount of dietary TRP for growth—in fact, alterations to dietary TRP has resulted in osmotic-based acute stress [98,133,134], where TRP supplementation in non-stressed fish has resulted in increased plasma cortisol levels, while the opposite effect has been observed in stressed groups [13,85]. Moreover, exogenous TRP supplementation varies depending on fish species and size, as well as exposure to stressful environments (i.e., crowding, high stocking density, pollutants, etc.) [98,134,135,136]. 

Oxidative stress (OS) is another mechanism by which cell and tissue damage can occur. OS is an increase between the production and accumulation of ROS, which are generated as by-products of oxygen metabolism in processes including immune activation, apoptosis, cell differentiation, and protein phosphorylation, as well as through exposure to environmental stressors and xenobiotics [137]. The consequences of increased cell exposure to ROS include reduced levels of adenosine triphosphate (ATP), lipid peroxidation, cell membrane depolarization, morphological changes in cell surfaces, and DNA damage [138]. Cellular damage marked by OS includes lipid peroxidation, increased levels of protein carbonyl, and decreased levels of antioxidant enzymes (SOD, catalase [CAT] and glutathione peroxidase [GPx]) [137]. The TRP-KYN pathway plays a role in the onset of OS (Reviewed in: [139]) and it has been proposed that TRP exhibits antioxidant activity as it reacts with free radicals and modulates antioxidant enzyme activities. In fact, there is accumulating evidence to support the use of TRP and its metabolites as antioxidants [140]. In rabbits, dietary TRP supplementation was effective in protecting against free radical generation and lipid oxidative damage produced by hypoxic myocardial injury [141]. Similarly, in weaned piglets, increasing dietary TRP levels resulted in enhanced antioxidant capacity (SOD and GPx), which attenuated the OS response induced by diquat injection [142]. In line with the work done in mammals, dietary TRP supplementation in ducks alleviated stress and improved growth performance and antioxidant activity (GPx and CAT). Similarly, (Reviewed In: [143,144]) it has been shown in fish that while increased supplementation of dietary TRP increases total SOD, CAT, glutathione activity, and total antioxidant capacity, an optimal range of TRP supplementation, is dependent on species [145].

Some TRP metabolites including 5-hydroxytryptophan (5-HTP), indole-3-acetic acid (IAA), 3-HAA, 3-HK and XA can act as ROS scavengers and modulate antioxidant enzymes. Interestingly, other KYN metabolites including QUIN, 3-HK and 3-HAA can generate oxygen radicals and ROS, thereby modulating OS [146,147,148]. Accordingly, the pharmacological inhibition of the KYN pathway can decrease levels of OS. For example, in a rat model of schizophrenia, inhibitors of TDO, IDO and KMO were able to prevent lipid peroxidation, decrease protein carbonyl levels, and increase SOD levels and catalase activity [149]. There is also considerable evidence linking aging with OS; in many aging-associated disorders, there are consistent reports of upregulated IDO (Reviewed in: [150,151]). Currently, reliable markers of OS include carbonylated proteins, malondialdehyde, 4-hydroxy-2-nonenal, and F2-isoprostanes analyzed by LC-MS/MS [152]. However, given that TRP and KYN metabolite levels have been repeatedly shown to be sensitive and specific to antioxidant capacity and associated gene changes, TRP-KYN levels, or varying ratios of the metabolites (i.e., 3-HAA/AA), may be surrogate markers for the organism’s response to insults causing increased ROS.

### 3.3. Environmental Contaminants

Environmental stressors are factors that constrain the productivity, survival, and reproductive success of organisms. These stressors can range from biological or social stress (i.e., predation, competition, and disease); physical disturbances to the landscape (i.e., weather events like tsunamis, hurricanes, volcanic eruptions, and wildfires, or anthropogenic deforestation, machinery trampling, and hikers); and chemical pollution (i.e., pesticides, flame retardants, personal care products, waste chemicals from industrial development, and air pollution). Increasingly, stressors due to anthropogenic activities have become the most critical influence on species and ecosystems [153]. Rapid industrialization and urbanization can dramatically change the composition and diversity of biotic communities. Alongside urbanization comes the increased distribution of environmental toxins through human activity. Some of these contaminants include heavy metals; polycyclic aromatic compounds (PACs) which include benzo[a]pyrene (BaP), polycyclic aromatic hydrocarbons (PAHs) and their heterocyclic, alkylated, halogenated, oxygenated, sulphated and nitrated analogs (Reviewed In: [154]); dioxins, including 2,3,7,8-tetrachlorodibenzo-p-dioxin (TCDD), polychlorinated dibenzofurans (PCDFs), polychlorinated dibenzo-p-dioxins (PCDDs); organochlorine compounds including polychlorinated biphenyls (PCBs), hexachlorocyclohexane isomers, dichlorodiphenyltrichloroethane (DDT) compounds, and hexachlorobenzenes (Reviewed In: [155]). Many of these contaminants can be released into the environment from natural and anthropogenic sources and have become ubiquitous in recent years. For example, PACs can be found in sea ice in the Arctic, industrialized harbors, the oil sands region in Canada, and major oil spills (e.g., the Exxon Valdez oil spill, Alaska; Prestige oil spill, Spain; Deepwater Horizon oil spill, Gulf of Mexico) [154,156]. Similarly, exposure to dioxins and furans can occur naturally through wildfires and through industrial processes such as waste incineration, the burning of fuels (i.e., wood, coal, and oil), and accidental/residential/structural fires. Other contaminants such as pesticides have an even greater public health concern; while they are designed to kill specific plants and insects, they often have harmful effects on non-target species. In mammals, exposure to pesticides has been linked to cancer, neurotoxicity and immunotoxicity [157]; endocrine disruption [158], reproductive effects, and birth and developmental defects (Reviewed In: [159]), and changes in energy homeostatic organs such as the liver and adipose tissue [160]. Similarly, in birds, organophosphates, neonicotinoids [161], carbamates and second-generation insecticides can result in decreased food consumption, weight loss, delayed migration, and decreases in the production, fertility, and hatchability of eggs (Reviewed In: [157,162]). Fish and other aquatic organisms are the most exposed in their environments due to the runoff of pesticides [163,164,165], oil spills [166,167,168,169], wastewaters from sewage containing personal care products [170,171,172,173], and chemical plant disposals; and have been extensively reviewed. Animal and human exposure to these chemicals can occur directly through pollution, or indirectly through food chain effects [174,175,176]. 

Many physiological homeostatic responses to environmental contaminants can be attributed to perturbations in pathways associated with GC signaling, inflammation, OS and, ultimately, alterations in TRP metabolism (Figure 6). For example, rats exposed to pyrethroid pesticides (deltamethrin and fenpropathrin) had significantly lowered KYNA production in cortical brain slices [177]. In chicken embryos, exposure to organophosphate and methylcarbamate insecticides was associated with lowered embryo NAD^+^ levels [178]; this effect was attributed to the inhibition of KYN formamidase, which impairs conversion of TRP to essential pyridine nucleotide cofactors. The majority of research in this area has focused on the link between environmental contaminants that act as AhR ligands and perturbations in the TRP-KYN pathway. Many environmental contaminants, particularly PACs, are known to act as AhR ligands; these compounds are also known to regulate IDO and TDO expression, thereby affecting the production of immunomodulatory TRP metabolites. For example, in mammals, it is well established that IDO expression and activity can be induced by AhR ligands including TCDD, BaP, and several PAHs [179,180]. Interestingly, KYN metabolites, including L-KYN, KYNA, XA, cinnabarinic acid, indigo, indirubin, and ultraviolet (UV) photoproducts of TRP such as 6-formylindolo[3,2-b]carbazole (FICZ) can act as AhR ligands and induce AhR target genes (CYP1A1, CYP1A2, and CYP1B1) (Reviewed In: [181]). Novikov et al. (2016) investigated the role of TRP-derived metabolites within malignant and non-malignant breast cancer cell lines and showed that (1) cell lines that expressed TDO produced sufficient intracellular KYN and XA concentrations to activate the AhR; (2) TDO overexpression led to excess KYN and XA, which accelerated the migration of tumour cells in an AhR-dependent manner; and (3) the environmental ligands TCDD and BaP, as well as the endogenous TRP-derivative FICZ mimic this effect [182]. Furthermore, AhR knockdown or inhibition significantly reduced TDO2 expression. KYN has also been shown to activate AhR in an autocrine/paracrine fashion, resulting in the suppression of antitumour immune responses and the promotion of tumour cell survival and motility [179]. Furthermore, FICZ—the UV photoproduct of TRP—binds to AhR with the highest affinity known to date of endogenous ligands; at high concentrations, FICZ behaves similarly to TCDD, exhibiting toxicity in fish and bird embryos, and playing a role in immunosuppression [183]. In birds, including the chicken (*Gallus gallus domesticus),* ring-necked pheasant (*Phasianus colchicus*), Japanese quail, and common tern (*Sterna hirundo*), FICZ has been identified as an avian AhR ligand [184], However, with the exception of changes in the avian CYP1A gene, other pathways mediated by AhR remain to be studied. 2-(1′H-indole-3′-carbonyl)-thiazole-4-carboxylic acid methyl ester (ITE)—a dietary TRP derivative—was discovered in porcine lung tissues and has been shown to be a high-affinity AhR ligand in humans, mice, and fish [185]. Taken together, these data suggest that across vertebrates, environmental exposure to certain contaminants can result in altered TRP-KYN metabolites that can contribute to a range of adverse health outcomes.

## 4. Conclusions

Many physiological responses to environmental contaminants can be attributed to perturbations in pathways associated with GC signaling, inflammation, oxidative stress, and, ultimately, alterations in TRP metabolism. Notably, environmental contaminants and their responses in vertebrate species are strikingly similar. Given that AhR and TRP metabolism are evolutionarily conserved across vertebrates (i.e., mammals, birds, fish), the crosstalk between xenobiotic receptors such as AhR, IDO/TDO immunoregulatory pathways, and altered stress indicators (i.e., antioxidant levels, ROS levels, and cortisol) suggest that alterations in TRP-KYN metabolism, metabolite levels, and ratio can be a cross-species marker of environmental exposure to chemical contaminants. Moreover, as ecotoxicological assessments are slowly moving away from evaluating the effects of a single compound to complex environmental mixtures, the TRP-KYN pathway provides a promising avenue to model the impacts of exposure to complex mixtures. Given its role in other biological processes, the TRP-KYN pathway provides many integrative biomarkers; both enzymatic ratios and metabolite levels can link environmental exposures to animal health, and broadly, ecosystem health. These integrative markers can be used as part of environmental effects monitoring (EEM) programs, where a few biomarkers could be monitored over a larger spatial area, across multiple species, trigger “investigation of cause” (IOC) studies or lead to focused studies to aid in the identification of mixtures and environmental factors that cause sublethal effects over acute and chronic time points.

## Figures and Tables

**Figure 1 ijms-23-06300-f001:**
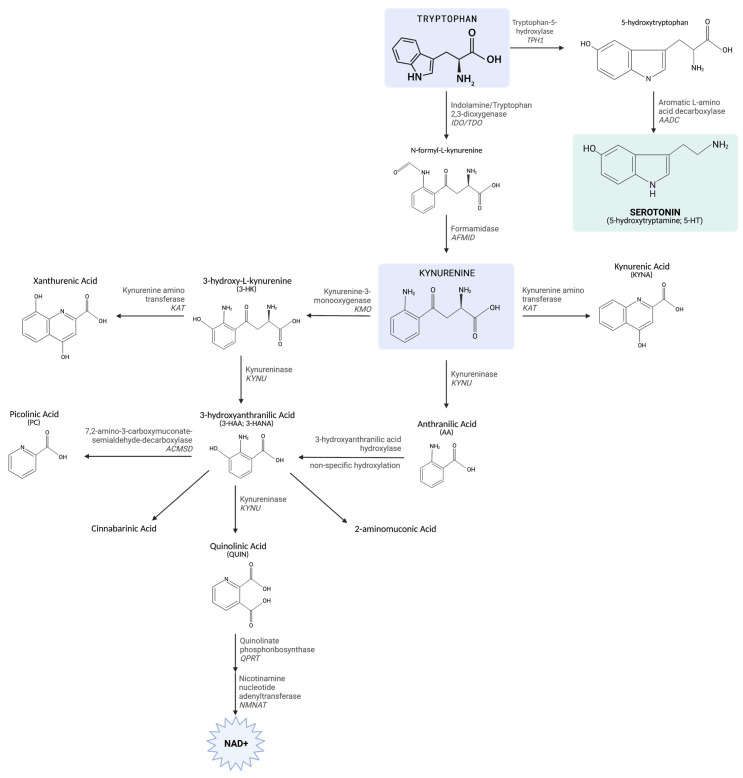
Key enzymes regulating TRP metabolism. The pathway depicting key regulators and targets that play a role on the TRP metabolism with a connection to KYN and 5-HT. Figure created on biorender.com.

**Figure 2 ijms-23-06300-f002:**
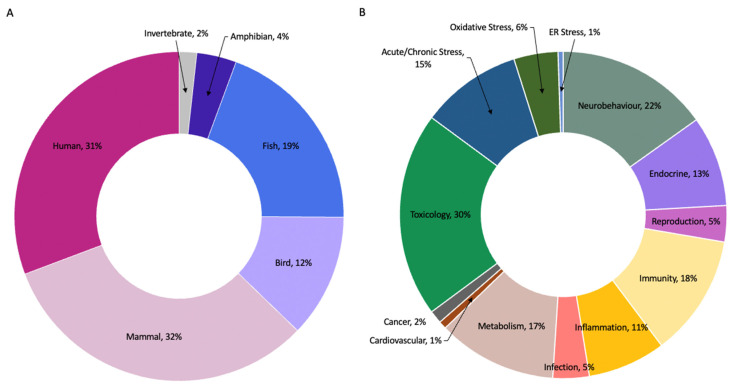
A review of the literature by (**A**) species and (**B**) physiological processes/pathways.

**Figure 3 ijms-23-06300-f003:**
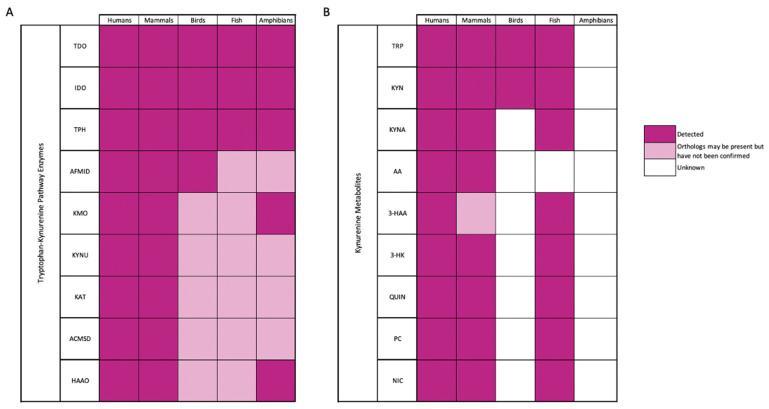
A review of kynurenine pathway (**A**) enzymes and (**B**) metabolites detected across animal kingdoms. If both gene and protein data was available, it was considered detected, and labelled dark pink. If enzymatic and metabolite orthologs may be present (via. computational curation of databases) but have not yet been confirmed in gene or protein levels, it was labelled light pink. If no information could be found yet, it was considered unknown and labelled white.

**Figure 4 ijms-23-06300-f004:**
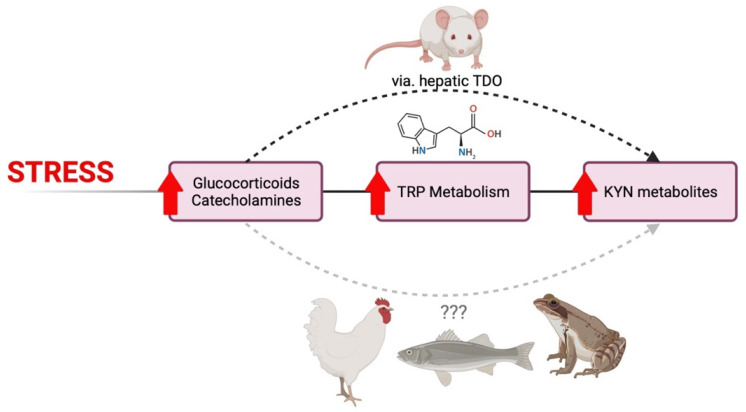
Across species, mammals, birds, and fish have similar responses to stress induced changes in TRP-KYN metabolism. Stress responses include increased GC and catecholamine signaling, which elevate TRP metabolism through the KYN pathway, as indicated by the levels of downstream metabolites. While it is generally accepted that this occurs through hepatic TDO induction in mammals, the molecular mechanism is not yet clear in other vertebrate species. Figure created on biorender.com.

**Figure 5 ijms-23-06300-f005:**
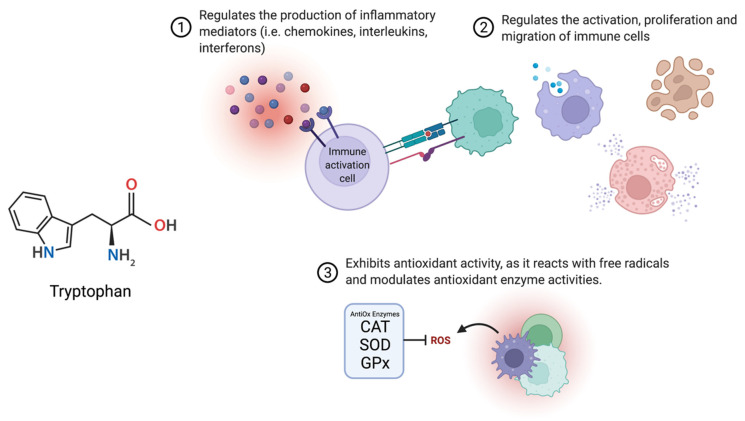
TRP functions as a regulator of inflammatory signaling molecules that control the immune response and the antioxidant response to cell stress and tissue damage. Figure created on biorender.com.

**Figure 6 ijms-23-06300-f006:**
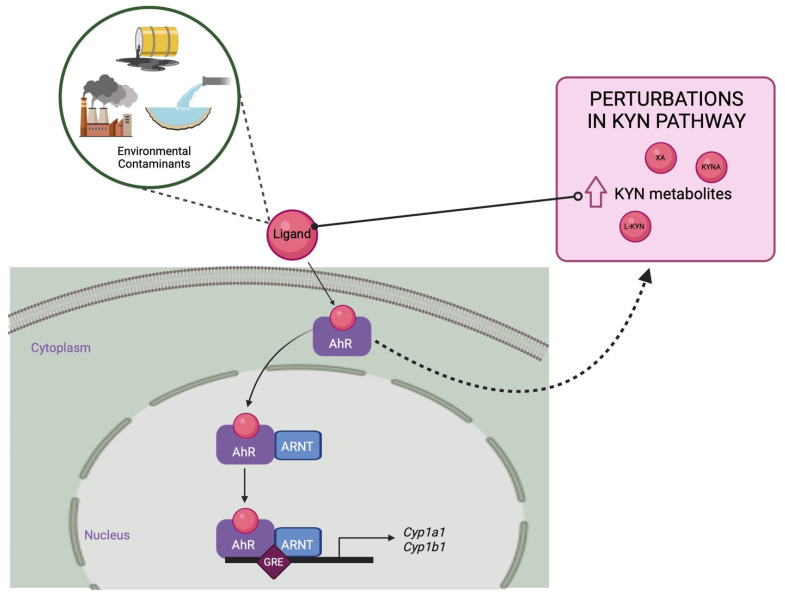
Alterations in TRP metabolism and KYN metabolites may be an indication of exposure to environmental contaminants. Many environmental contaminants (i.e., PACs) are known to act as AhR ligands and are also known to regulate IDO and TDO expression, thereby affecting the production of immunomodulatory KYN metabolites. Moreover, many of these metabolites can also act as AhR ligands and induce AhR target genes. Figure created on biorender.com.

## Data Availability

Not applicable.

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
