# Peer review of "An Emerging Cross-Species Marker for Organismal Health: Tryptophan-Kynurenine Pathway"

_ijms, 2022, doi:10.3390/ijms23116300_

Round 1

Reviewer 1 Report

The review focuses on key pathways affecting TRP metabolism  in vertebrates and analyses the consequences of altered TRP metabolism in mammals, birds, amphibians, and fishes. More importantly it suggest that Tryptophan-Kynurenine (TRP-KYN) metabolism  can be used as biomarker of the ecosystem health. The manuscript is a significant contribution for the interdisciplinary scientific community involved. It is well written by experts in a very specific field and the various topics are clearly and consistently presented. It is easy for the reader going through the paper and following the discussion.

Specific point

-Methods.In this chapter, the authors describe how the literature search was conducted.The strategy that has been used does not contribute to the impact of the work and can be deleted as well as the corresponding figures 2 and 3.

Reviewer 2 Report

9 April 2022

Regarding the review of manuscript “An Emerging Cross-Species Marker for Organismal Health: Tryptophan-Kynurenine Pathway” by Jamshed L et al., submitted to International Journal of Molecular Sciences (IJMS)

Manuscript ID: ijms-1637473

Dear Authors,

The tryptophan(Trp)-kynurenine (KYN) metabolic pathway is a major route of Trp catabolism, which is influenced by pyschological, biological, and environmental stress. However, little is known about its role in response to enviromental contaminats. In the present review entitled ‘An Emerging Cross-Species Marker for Organismal Health: Tryptophan-Kynurenine Pathway’, Jamshed and colleagues discussed key pathways affecting Trp-KYN metabolism in vertebrates, highlighting the consequences of altered tryptophan metabolism in some spieces. For this purpose, the authors further discussed its implication in environmental toxicology, since the metabolic system is conserved across vertebrates, concluding that Trp-KYN metabolites are potential candidates for organismal as well as ecosystem healthy. The main strength of this original review article is that it addresses an interesting and timely question, highlighting the current understanding of the Trp-KYN system and its potential use for biomarkers. In general, I think the idea of this article is really interesting and the authors’ fascinating observations on this timely topic may be of interest to the readers of the International Journal of Molecular Sciences. However, some comments, as well as some crucial evidence that should be included to support the author’s argumentation, needed to be addressed to improve the quality of the manuscript, its adequacy, and its readability before its publication in the present form, in particular reshaping parts of the Introduction, Methods, Results and Discussion sections by adding more evidence and theoretical constructs.

Please consider the following comments:

  1. Abstract: Please proportionally present background, rationale, purpose, and conclusion.
  2. Keywords: Please reduce the number of keywords up to ten.
  3. Introduction: I suggest reorganizing the introduction section and updating information to present a current perspective view on the Trp-KYN metabolic system including its enzymes and metabolites. The descriptions provided by the authors are inaccurate and may lead to misunderstanding of the biosystem. For example, there are isoforms of indoleamine-2,3-dioxygenase and L-kynurenine aminotransferase. The redox activity of KYN metabolites depends on the environment and the concentration. KYN is antioxidants, Kynurenic acid has been repeatedly documented as neuroprotective; however, the higher concentration is considered to to cause cognitive impairment in schizophrenia, but a lower dose contributes to memory enhancement. Thus, the biological roles of KYN metabolites remained inconclusive and many of them possess multiple and dual properties. Furthermore, the potential use of the concentration and the ratio KYN metabolite for various biomarkers in neurological and mental illnesses has been discussed recently. The section should also describe other bioactive properties of KYN metabolites such as G protein-coupled receptor 35 and aryl hydrocarbon receptor. Probably, an independent section for the Try-KYN metabolic system may help introduce the complex and relatively less known biosystem under extensive research, after the introduction describing the solid rationale which is crucial for this paper pioneering in the field and the purpose of this review paper in the end.
  4. Methods: Please clearly state this manuscript is a narrative, scoping, or systematic review. This must be declared in the abstract and in the end of the introduction and preferably stated in the title if it is a scoping or systematic review. If it is a narrative review, the methods section is unnecessary. If it is a scoping or systematic review, necessary element must be presented such as PRISMA flow chart and/or risk of bias assessment. If it is scoping review, please state a gap to systematic review is preferably. The search criteria are described in detail and the search results are well presented, indeed. I recommend presenting a scoping review with clearly describing the database used, search performer with initials, and PRISMA chart. Finally, the search results should be presented in an independent section of Results.
  5. Discussion: The section is described in detail, but the lack of introductory part makes it less informative and thus, it fails to attract more readers, even though this section is the most important part of the whole manuscript in which the authors can present their expertise. I recommend presenting a logical sequence of the Discussion in the beginning of the section, leading to the following subsections. In my opinion, this research article would be more compelling and useful to a broad readership if the authors moved beyond and discussed theoretical and methodological avenues in need of refinement, using this evidence to suggest a path forward. This may include potential of this review article, weakness and limitation, technology required to advance in the field, and future research direction.
  6. In my opinion, I think the ‘Conclusions’ paragraph would benefit from some thoughts as well as in-depth considerations by the authors because as it stands, it is very descriptive but not enough theoretical as a discussion should be. The authors should make an effort to explain the theoretical implication and the translational application of their research.
  7. References: According to the Journal’s guidelines, authors should consider revising the bibliography, as there are several incorrect citations. Indeed, according to the Journal’s guidelines, they should provide the abbreviated journal name in italics, the year of publication in bold, the volume number in italics for all the references. Also, they should have provided the DOI number for each reference. I also recommend updating the references, as they are relatively outdated.

Overall, the manuscript contains six figures, no table and 182 references. I believe that the manuscript may carry important value presenting that monitoring the Trp-KYN metabolic system is of potential use for biomarkers in cross-species and thus environmental health. I hope that, after these careful revisions, the manuscript can meet the Journal’s high standards for publication. I am available for a new round of revision of this review.

Best regards,

Reviewer

Round 2

Reviewer 2 Report

13 May 2022

Regarding the 2nd review of manuscript “An Emerging Cross-Species Marker for Organismal Health: Tryptophan-Kynurenine Pathway” by Jamshed L et al., submitted to International Journal of Molecular Sciences (IJMS)

Manuscript ID: ijms-1637473

Dear Authors,

In the present review entitled ‘An Emerging Cross-Species Marker for Organismal Health: Tryptophan-Kynurenine Pathway’, Jamshed and colleagues discussed key pathways affecting Trp-KYN metabolism in vertebrates, highlighting the consequences of altered tryptophan metabolism in some spieces.

I appreciated the Authors' answers to the points that I raised in the first round of review, as well as their clarifications of some of my concerns. However, despite my suggestions to provide more information, the authors need to add some crucial studies that could have allowed to enrich and complete the theoretical framework.

First of all, I recommend expanding the abstract to 200 words, clearly filling the gaps between the importance of tryptophan (Trp)-kynurenine (KYN) in organisms and implications for environmental and wildlife toxicology and between the aims and the postulate the authors declared.

In my opinion, I still have hard time to remove my impression that the description of the Try-KYN system in this manuscript remained off-focused and thus, I certainly believe that adding more findings from the recently published studies that have focused on the current understanding of the Trp-KYN metabolic system. For example, the authors hastily summarized the activities of KYN metabolites; however, I recommend clearly presenting the previous concept and emerging evidence including the chemical properties, the physiological roles, and potential biomarkers of KYN metabolites (https://doi.org/10.3390/biomedicines10040849; https://doi.org/10.3390/antiox11010031; doi: 10.1016/j.neubiorev.2020.08.010). I believe this discussion may serve as a crucial argument based on the current understanding, leading to the potential use of the metabolic system for organismal and ecosystem scales, which the authors intend to present in this manuscript (doi: 10.17219/acem/139572). The authors listed only three neurological and psychiatric disorders for the alteration of the metabolic system. This may mislead general readers who are not familiar with the Try-KYN pathway (https://www.mdpi.com/2227-9059/9/8/897; doi: 10.1016/j.pharmthera.2021.107807; https://www.mdpi.com/1422-0067/22/20/11055; https://www.mdpi.com/1422-0067/22/18/10134; doi: 10.3390/ijms21176045).

Overall, the manuscript contains six figures, no table and 186 references. I believe that the manuscript may carry important value presenting that monitoring the Trp-KYN metabolic system is of potential use for biomarkers in cross-species and thus environmental health. I hope that, after these careful revisions, the manuscript can meet the Journal’s high standards for publication. I am available for a new round of revision of this review.

Best regards,

Reviewer

Round 3

Reviewer 2 Report

27 May 2022

Regarding the 3rd review of manuscript “An Emerging Cross-Species Marker for Organismal Health: Tryptophan-Kynurenine Pathway” by Jamshed L et al., submitted to International Journal of Molecular Sciences (IJMS)

Manuscript ID: ijms-1637473

Dear Authors,

The authors did an excellent work clarifying the questions I have raised in the previous round of review. Currently, this paper is a well-written, timely piece of research and provides a useful study addressing an interesting and innovative question, presenting that monitoring the tryptophan-kynurenine metabolic system is of potential use for biomarkers in cross-species and thus environmental health. Overall, this is a timely and needed work, thus I believe that manuscript now meets the Journal’s standards for publication. I am always available for other reviews of such interesting and important articles. I look forward to seeing further study on this issue by these authors in the future.

Thank you for your work.

Best regards,

Reviewer